# Real-Time Analysis of Hand Gesture Recognition with Temporal Convolutional Networks [note 1]

**DOI:** 10.3390/s22051694

**Published:** 2022-02-22

**Authors:** Panagiotis Tsinganos, Bart Jansen, Jan Cornelis, Athanassios Skodras

**Affiliations:** 1Department of Electrical and Computer Engineering, University of Patras, 265 04 Patras, Greece; skodras@upatras.gr; 2Department of Electronics and Informatics (ETRO), Vrije Universiteit Brussel, 1050 Ixelles, Belgium; bjansen@etrovub.be (B.J.); jpcornel@etrovub.be (J.C.); 3Imec, Kapeldreef 75, 3001 Leuven, Belgium

**Keywords:** sEMG, hand gesture recognition, deep learning, CNN, TCN, real time, attention

## Abstract

In recent years, the successful application of Deep Learning methods to classification problems has had a huge impact in many domains. (1) Background: In biomedical engineering, the problem of gesture recognition based on electromyography is often addressed as an image classification problem using Convolutional Neural Networks. Recently, a specific class of these models called Temporal Convolutional Networks (TCNs) has been successfully applied to this task. (2) Methods: In this paper, we approach electromyography-based hand gesture recognition as a sequence classification problem using TCNs. Specifically, we investigate the real-time behavior of our previous TCN model by performing a simulation experiment on a recorded sEMG dataset. (3) Results: The proposed network trained with data augmentation yields a small improvement in accuracy compared to our existing model. However, the classification accuracy is decreased in the real-time evaluation, showing that the proposed TCN architecture is not suitable for such applications. (4) Conclusions: The real-time analysis helps in understanding the limitations of the model and exploring new ways to improve its performance.

## 1. Introduction

Accurate gesture recognition is important for a number of applications, including human computer interaction [1], prosthesis control [2], and rehabilitation gaming [3,4]. Surface electromyography (sEMG) signals measured from the forearm contain useful information for decoding muscle activity and hand motion.

Machine Learning (ML) classifiers have been used extensively for determining the type of hand motion from sEMG data. A complete pattern recognition system based on ML includes acquiring data, extracting features, specifying a model, and reasoning about new data. In the case of gesture recognition based on sEMG, electrodes attached to the arm and/or forearm acquire the EMG signals, and the typical extracted features are RMS, variance, zero crossings, and frequency coefficients that are applied as inputs to classifiers such as k-NN, SVM, MLP, and Random Forests [5].

Recently, Deep Learning (DL) models have been successfully applied to sEMG-based gesture recognition. In these approaches, EMG data are represented as images, and a Convolutional Neural Network (CNN) is used to determine the type of gesture. Although EMG signals are time-series data, to our knowledge, only recently appropriate DL models (e.g., Recurrent Neural Network—RNN, such as Long Short-Term Memory—LSTM) have been utilized [6,7,8]. In this work, taking into account the outcomes of [9], we investigate the application of temporal convolutional networks (TCN) [10] to the problem of sEMG-based gesture recognition. In contrast to the image classification approach, EMG signals are considered as a multi-dimensional time series, and only 1D convolutions are applied. Additionally, the outputs of the convolutions are computed using only past and current data (causal convolutions).

The problem of sEMG-based hand gesture recognition has been studied thoroughly using either conventional ML techniques or DL methods. In the case of ML-based methods, the first significant study is presented in [11] for the classification of four hand gestures using time-domain features extracted from sEMG measured with two electrodes. The authors of [12] achieve a 97% accuracy in the classification of three grasp motions using the RMS value from seven electrodes as the input to an SVM classifier. The works of [13,14,15] evaluate a wide range of EMG features with various classifiers for the recognition of 52 gestures (Ninapro dataset [16]). The best performance is observed with a combination of features and a Random Forest classifier, resulting in 75% accuracy.

On the other hand, the first DL-based architecture was proposed in [17]. The authors built a CNN-based model for the classification of six common hand movements, resulting in a better classification accuracy compared to SVM. In [18], a simple model consisting of five blocks of convolutional and average pooling layers resulted in accuracy figures comparable, though not higher, to what was obtained with classical ML methods. In our previous work [19], we have investigated methods to improve the performance of this basic model. The results have shown that opting for max pooling and inserting dropout [20] layers after the convolutions boosts the accuracy by 3% (from 67% to 70%). The works of [21,22] incorporate dropout and batch normalization [23] techniques in their methodology. Apart from differences in model architectures, they measure EMG signals using a high-density electrode array, which has been proven effective to myoelectric control problems [24,25,26]. Using instantaneous EMG images, the authors of [21] achieve 89% accuracy on a set of eight movements, whereas in [22], a multi-stream CNN architecture is employed, resulting in 85% accuracy on the Ninapro dataset. A quick comparison of the state-of-the-art accuracy is difficult to make because of the lack of standardization of the databases and the number of gestures to be recognized [27] (p. 36).

Other important works based on DL architectures deal with the problem of model adaptation. In [28], the technique of adaptive batch normalization (AdaBN) [29] updates only the normalization parameters of a pretrained model, whereas in [30], the authors use weighted connections between a source network and the model instantiated for a new subject. Additionally, in [30], they propose data augmentation methods for sEMG signals.

The current article is based on the conference presentation [31], where the offline analysis of the TCN models is described. To extend that work, a real-time simulation is performed using an existing sEMG database that allows the analysis of the performance of the TCN models for real-time application purposes. In addition, the optimization procedure was redesigned in order to incorporate the findings of our previous study [32]. The main contributions presented in this paper are as follows:Analysis of the real-time performance of the proposed TCN models using a simulation experiment;Improved offline accuracy compared to our previous study [31] as a result of the optimized hyperparameter values.

The paper is organized as follows. In Section 2, we give a detailed description of the proposed TCN architecture and the experimentation methodology. Section 3 provides the results followed by a discussion. Finally, Section 4 summarizes the conclusions.

## 2. Materials and Methods

In the domain of sEMG-based gesture recognition, the majority of existing works solve this problem as an image classification task, where recordings of multiple electrodes are arranged in a 2D representation that resembles a gray-scale image. In this study, we continue our exploration of a time sequence recognition model and specifically the TCN model, which was originally described in detail in our previous publication [31]. Here, we provide a brief description of the TCN model as well as more implementation details. The main characteristic of TCNs is the use of causal convolutions, which given an input sequence of length *N* produce an output sequence of the same length. A potential limitation of using convolutional networks for a time-sequence problem could be the lack of modeling long-term correlations between the samples of the signal, which LSTMs solve with the state vector. However, TCN models can solve this problem by using dilated convolutions that enable a large receptive field (RF) [33] that may cover the entire duration of the signal. Furthermore, residual connections [34] allow training deeper networks, which can extract better features for the classification of the signals. Considering that our task is to classify sEMG signals, the output of the model should be a single class label characterizing the input sequence. Thus, the last convolutional layer of TCN is further processed by either an average over time (AoT) calculation or an attention (Att) mechanism [35] followed by a softmax activation.

An sEMG signal is represented as a *C*-dimensional sequence of length *N*, $x=\{x_{0},\cdots ,x_{N}\}$, where $x_{i}\in R^{C}$ and $x\in R^{N\times C}$. The output feature map of a causal convolutional layer of *K* filters is a sequence $y\in R^{N\times K}$ where each element of the first dimension is: (1)$$y_{n}=f\left(x_{0,\cdots ,n}\right),\;\forall n<N$$

Specifically for dilated convolutions:(2)$$y_{n}={\left(x{*}_{d}h\right)}_{n}=\sum_{i=0}^{2p}x_{n-di}h_{i}$$
where ${*}_{d}$ is the operator for dilated convolutions, *d* is the dilation factor, and *h* is the filter’s impulse response of length 2p+1. For the (l+1)-th convolutional layer of $K^{l+1}$ filters, the output $y^{l+1}\in R^{N\times K^{l+1}}$ is computed as:(3)$$y_{n,k}^{l+1}=\sum_{j=0}^{K^{l}-1}\sum_{i=0}^{2p}y_{n-di,j}^{l}h_{i,j,k}\;\forall n<N$$
where $k\in [0,K^{l+1}-1]$, $y^{l}\in R^{N\times K^{l}}$ is the output of the previous layer or the input sequence x if l=0, i.e., $y^{0}=x$. A schematic representation is shown in Figure 1a.

The architecture consists of successive residual blocks (Figure 1b) where the output feature map is the element-wise summation of the input feature map and the same input feature map processed by a series of causal convolutional and dropout layers. After every convolutional layer, the value of the dilation ratio is doubled, i.e., $d^{l+1}=2d^{l}$. The RF after a convolutional layer is computed as:(4)$${\mathrm{RF}}^{l+1}={\mathrm{RF}}^{l}+2pd^{l+1},\;l\in \left[0,\cdots ,L\right]$$
where l=0 is the input layer, $RF^{0}=0$, and $d^{1}=1$.

In a TCN model that consists of *L* convolutional layers, the output of the last layer, $y^{L}$, is used for the sequence classification assigning one out of *G* gesture labels. In the case of the AoT approach, the class label $\hat{o}$ attributed to the sequence is found through a fully connected layer with softmax activation function: (5)$$\begin{array}{ccc} s & = & \frac{1}{N}\sum_{n=0}^{N-1}y_{n}^{L} \end{array}$$(6)$$\begin{array}{ccc} \hat{o} & = & \mathrm{softmax}(s\cdot W_{o}+b_{o}) \end{array}$$
where $s\in R^{1\times K^{L}}$ and $W_{o}\in R^{K^{L}\times G}$, $b_{o}\in R^{1\times G}$ are trainable parameters. Otherwise, when using the Att mechanism, the class label is calculated as follows [35]:(7)$$\begin{array}{ccc} v & = & \tan h(y^{L}\cdot W_{a}+b_{a}) \end{array}$$(8)$$\begin{array}{ccc} a & = & \mathrm{softmax}(v\cdot u_{a}) \end{array}$$(9)$$\begin{array}{ccc} s & = & \sum a\circ y^{L}=\sum_{n=0}^{N-1}a_{n}y_{n}^{L} \end{array}$$(10)$$\begin{array}{ccc} \hat{o} & = & \mathrm{softmax}(s\cdot W_{o}+b_{o}) \end{array}$$
where $W_{a}\in R^{K^{L}\times K^{L}}$, $b_{a}\in R^{1\times K^{L}}$ are trainable parameters that compute a hidden representation $v\in R^{N\times K^{L}}$ from the TCN output, $u_{a}\in R^{K^{L}\times 1}$ is a learnable context vector, $a\in R^{N\times 1}$ is the vector of normalized importance weights, ∘ denotes the element-wise multiplication, and $s\in R^{1\times K^{L}}$ is the weighted sum across the time steps *n* of $y^{L}$ based on the importance weights. The output label $\hat{o}$ is calculated as in AoT through the softmax activation.

The proposed TCN architecture was evaluated on data from the first dataset of the Ninapro database. It includes data acquisitions of 27 healthy subjects (7 females and 20 males of age 28 ± 3.4 years) that perform each of the 52 gestures 10 times (repetition sequences). The types of gestures can be divided into three groups: (i) basic finger movements, e.g., index flexion/extension, (ii) isometric, isotonic hand configurations and basic wrist movements, e.g., fingers flexed together in fist, wrist flexion, and (iii) grasping and functional movements, e.g., prismatic pinch grasp, large diameter grasp. The data are acquired with 10 electrodes (OttoBock MyoBock 13E200-50), of which eight are placed at equal distances around the forearm and the other two are placed on the main activity spots of the large flexor and extensor muscles (flexor and extensor digitorum superficialis respectively) of the forearm [14].

The experimentation is based on existing works that have used this dataset [18,19,22] and follow an intra-subject evaluation, where the train/test split of the data is performed for each subject separately based on gesture repetitions. Specifically, for each subject, a new model is created with a random initialization of the weights and trained on data from seven repetitions (i.e., 1, 3, 4, 6, 8, 9, 10) and tested on the remaining three (i.e., 2, 5, 7). As performance metrics, we use the top-1 and top-3 accuracies (i.e., the accuracy when the highest and any of the 3 highest output probabilities match the expected gesture) averaged over all the subjects. Considering that the idle gesture is overrepresented in the dataset, a weighted average of the metrics is calculated where the weight for each gesture label is inversely proportional to the number of samples of that gesture.

The experiments consist of four different versions of the TCN model. The first distinction is based on the size of the RF, where models with an RF equal to 300 ms (short) and 2500 ms (long) were implemented. These values were achieved by using an exponential dilation factor $d=2^{l}$ for the *l*-th layer in the network. Secondly, the classification was based on either the AoT or the Att mechanism described above. Therefore, four models were evaluated using complete gesture repetition sequences as input. The details of each model are shown in Table 1.

The optimization of the models was based on the procedure described in our previous study on TCNs [31]; however, the findings of [32] regarding data augmentation techniques for sEMG data were incorporated. To this effect, an extended hyperparameter search was performed, which resulted in the following. All networks were trained using the Adam optimizer [36] for 100 epochs with early stopping, a constant learning rate of 0.01, and a batch size of 128. To avoid overfitting the networks due to the small training set (size of 53×7=371), the training data of each subject were augmented by a factor of 10 using the Wavelet Decomposition (WD), Magnitude Warping (MW), and Gaussian Noise (GN) methods described in [32]. The details of the augmentation hyperparameters are shown in Table 2. Finally, dropout layers were appended after each convolution with a forget rate of 0.05. These values were selected after performing a grid search on a validation set of five randomly selected subjects.

This study aims at providing insights into the real-time performance of the TCN models. For this purpose, we performed a simulation experiment using the test data where the input to the models was given a single sample at a time. During training and in the offline evaluation, the signals that correspond to the entire gesture duration are used as input to the models. On the other hand, in the real-time experiment, the sEMG signals of the test set repetitions are provided as input in a sample-by-sample fashion in segments of 200 ms (20 samples for Ninapro DB1). For example, initially, only the first sample $x_{0}$, i.e., a vector of 10 values, is fed to the model, while at the *n*-th iteration, the input consists of a sequence of 20 vectors $x_{n-20,\cdots ,n}\in R^{20\times 10}$. The process is repeated until the entire signal of length *N* is covered. The label predictions of each model are recorded for further analysis. To utilize the parallel processing of the GPU, the input is zero-padded up to a maximum sequence length. Additionally, these zero values are masked so that they do not interfere during training and inference.

The model predictions during the real-time experiment are processed as follows. Firstly, we utilize the classifier of [37] to compute the real-time accuracy. This approach is similar to a majority voting classifier, but instead, it assigns the label of the gesture that is predicted the most times above a threshold in an analysis window *w*. To that effect, at any given iteration *n*, we count the amount of times each label *i* is predicted, $N_{i}$. If an sEMG signal is classified more than τ times with a label *l*, i.e., $N_{l}\geq \tau$, this label is assigned as the predicted gesture for the analysis window *w*. If the threshold is not met, a ‘no gesture’ label is assigned. Thus: (11)$$\Psi_{n}\left(\hat{o}\right)=\left\{\begin{array}{c} l,\;N_{l}\geq \tau \\ -1,\;\mathrm{otherwise} \end{array}\right.$$

Each sEMG sequence generates a sequence of gesture label predictions. For both types of TCN models, i.e., AoT and Att, the predicted label sequences are compared to the corresponding ground truth labels, and the index of the first correct model prediction as well as the first index for which $\Psi (\hat{o})\neq -1$ are recorded. Divided by the sampling rate of the sEMG signals of the Ninapro DB1 dataset, i.e., 100 Hz, these indices are mapped to the time in seconds that the model requires to successfully predict the performed gesture. Then, for the Att models, the timings are compared to the peak of the attention weights distribution computed during training.

The models were developed in Python using Tensorflow/Keras libraries. The training was performed on a workstation with an AMD Ryzen 9 3950X 16-Core Processor, 126 GB RAM, and an Nvidia GeForce RTX 3080, 11GB GPU. Each training epoch was completed in approximately 30 s.

## 3. Results and Discussion

This section provides the experimental results for the offline and real-time experiments. In the first case, the accuracy and loss curves along with the confusion matrices for each TCN model are described. For the real-time experiment, the distribution of ‘the timings until the first correct prediction’ for each gesture is shown in the form of a boxplot, while the correlation coefficients of the relationship between these timings and the peaks of the attention weights are presented.

Figure 2 shows the loss and accuracy curves on the training and validation sets, while in Figure 3, the confusion matrices on the test set are shown. The average attention weight distributions extracted from the training set are shown in Figure 4. Table 3 presents the offline and real-time classification accuracy metrics for each of the models.

For the classifier of Equation (Equation 11), the analysis window *w* and the label count threshold τ were set to 12 and 300 ms, respectively. The classification accuracy for different values of the two parameters is shown as a heatmap in Figure 5. Boxplots showing the distribution statistics of ‘the timings until the first correct prediction’ are presented in Figure 6, Figure 7, Figure 8 and Figure 9.

### 3.1. Offline Analysis

In the offline analysis, the entire sequence of each sEMG signal was used as input to the models. The training and validation loss curves (Figure 2) show that the models were trained until convergence, while the early stopping approach helped avoid overfitting. The range of standard deviation of the results is similar to what can be achieved from other CNN models applied to the Ninapro dataset. The test accuracy metrics (Table 3) were increased by approximately 2% compared to our previous implementation [31]. However, the difference is significant only for the AoT models (p-value < 0.05) and not in the case of attention-based ones. We assume that this improvement stems from the better augmentation methods utilized in this study. Furthermore, we observe that the accuracy of the attention-based models is slightly lower than that of the time average.

The error analysis based on the average confusion matrices (Figure 3) confirms our previous findings that some gestures are difficult to classify correctly [31]. All confusion matrices display a few clusters of misclassifications between the following gestures: (i) thumb adduction–abduction and flexion–extension (labels 9–12), (ii) thumb opposing little finger and abduction of all fingers (labels 16–17), (iii) power grip and hook grasp (labels 31–32), (iv) types of three finger grasps (labels 40–42), and (v) types of pinch grasps (labels 43–44).

For the model using the attention mechanism, the importance weights was extracted and the average weight distribution per gesture is shown in Figure 4. Each plot shows the average values of the attention weights for the duration of the training repetitions of each gesture. The y-axis is the amplitude of the weights, and the x-axis corresponds to the time in s. For the Att300 model, we can see that in most of the gestures, the peaks are between 2 and 4 s. This means that for the model, this region is the most important for the classification.

### 3.2. Real-Time Analysis

In the real-time analysis, segments of 20 samples (200 ms) of sEMG signals were used as input to the models. The time from the start of the gesture until the first correct prediction is shown as a boxplot, where the measurements are collected for each gesture repetition in the test set across the subjects in the dataset. Figure 6, Figure 7, Figure 8 and Figure 9 show the results for each model. In general, easy to classify gestures (label 1–12) have lower times compared to more complex grasps (labels 30–52). This means that the model requires a bigger amount of input samples in order to correctly classify a more difficult gesture. Furthermore, the models with larger RF require more time to make a correct prediction, while the amount of outliers is smaller in the attention-based models. This is expected, since during training, the calculation of the last layer feature maps on which the classification is based takes into account a larger portion of the input signal.

Next, the correlation between the time of the attention peak and the time to the first correct prediction is calculated for the Att300 and Att2500 models. Measurements from all the subjects and the gestures are taken, and then, the Pearson’s correlation coefficient is calculated. The results for the two models are shown in Figure 10. In both models, no linear relationship was found with r=−0.0579 (Att300) and r=−0.0216 (Att2500). This means that we cannot make any assumptions about the time until the first correct prediction by looking at the attention weights.

The output sequence of the class labels was further processed over an analysis window w=300 ms and a label prediction threshold τ=12 using Equation (Equation 11). The window was selected in order to conform to real-time requirements [38], while τ was selected such that the accuracy is maximized. In Figure 5, the real-time accuracy is shown for various values of these parameters. The accuracy is maximized for w=1.6 s and τ=60 with a value of 0.5660. As expected, increasing both the analysis window and the label prediction threshold improves the accuracy performance. However, the response time, i.e., the time from the start of the analysis window until a prediction is made, also increases. Figure 11 shows the classification accuracy and the response time for w=1.6 s while τ is varied from 10 to 80. The accuracy improves when the τ threshold increases, because the model can filter out wrong predictions that are repeated less frequently than the correct label within the processing window *w*. After a certain point, further increase of τ deteriorates the performance, since the count of the correct label does not meet the threshold and the model outputs either a different label or ‘no gesture’.

The above results highlight the difference between the offline and the real-time performance of the proposed models. These TCN architectures can successfully discriminate between a large number of gesture labels when the entire signals are fed as inputs. However, the real-time evaluation suggests that the specific architecture configuration is not suitable for real-time applications. A similar analysis as ours on the Ninapro DB1 dataset was not found. However, other approaches have been successful in Ninapro datasets. In [39], a TCN encoder–decoder network is trained to predict a gesture label at each timestep, i.e., the optimization procedure is oriented toward a real-time application. In Ninapro DB2, which contains 40 different gestures, the classification accuracy is 0.8270, and the average response time was calculated at 4.6 ms. In addition to making a fast correct prediction, the predictions were highly stable. Furthermore, in the work of [40], a TCN model consisting of three convolutional layers is trained on 150 ms window segments using sliding windows. The accuracy in classifying eight gestures in Ninapro DB6 was 0.7180 using only the steady portion of the signal; i.e., the transients at the beginning and the end of the signal were removed before the training.

Next, in order to take into account the real-time requirements into the training, the models were optimized using segments of the sEMG signal. Specifically, we used segments with a duration of 200 ms (20 samples), i.e., equal to the real-time experiment, based on the sliding windows method with a window size of 20 samples and step of 1 sample. The remaining hyperparameters, i.e., augmentation, model architecture, training epochs, learning rate, and batch size, were the same as before. Then, the models were evaluated using segments of 200 ms and processed further with an analysis window w=300 ms and τ=12. The results shown in Table 4 agree with the results found in the above-mentioned works that follow a similar approach for the training [40]. Comparing Table 3 and Table 4, we observe that in the latter, the real-time accuracy has been improved, while the response time remains in the same range of values as before. Although the average response time is below the real-time threshold of 300 ms, it is much higher than the response time achieved in [39].

## 4. Conclusions

This study follows our previous work regarding the application of a TCN model to the problem of sEMG-based recognition [31]. In contrast to existing works that address the problem as an image classification task, the proposed model can categorize complete sEMG sequences. The architecture consists of a stack of layers that perform temporal causal convolutions, while the class label is computed either with an average over time (AoT) or an attention mechanism (Att) method. The model was successful in classifying the sEMG signals of 53 gestures in Ninapro DB1 in the offline evaluation, i.e., when the entire signal was used as input to the model, and an augmentation scheme based on [32] further boosted the performance. However, in the real-time evaluation experiment consisting of providing the model with 200 ms segments, the classification accuracy was quite low. A comparison with similar approaches of TCN models in the literature suggests that a tailor-made data-feeding procedure during training adapted for real-time application can be more effective than our extension of an offline analysis scheme. Finally, using a training procedure based on sliding windows, the real-time performance improves as expected.

## Figures and Tables

**Figure 1 sensors-22-01694-f001:**
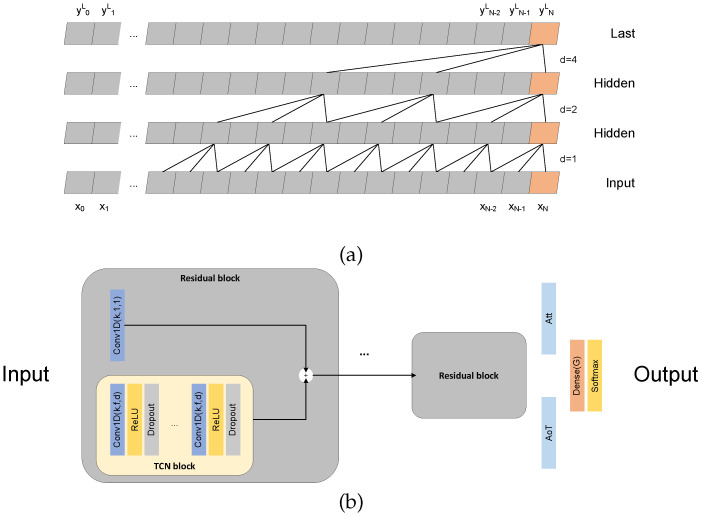
The TCN architecture. (**a**) Graphical representation of temporal causal convolutions with exponential dilation factor *d*. (**b**) Each residual block performs a summation of a causal convolution and the output of the TCN block that consists of a succession of causal convolutions, ReLU, and Dropout layers. The Conv1D layer hyperparameters are *k* number of filters, *f* filter size, and *d* dilation ratio.

**Figure 2 sensors-22-01694-f002:**
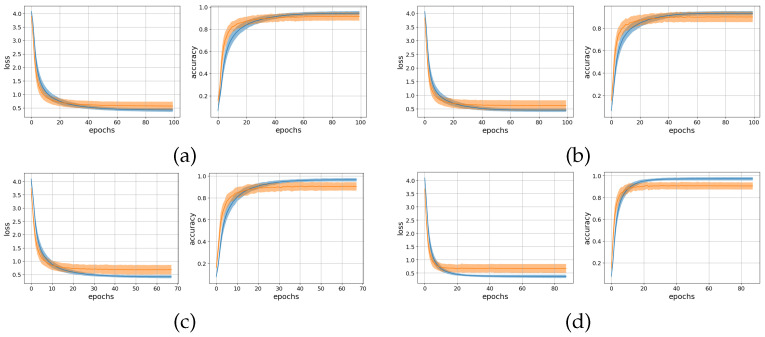
Loss and accuracy curves for training set (blue) and test set (orange) for each of the models. (**a**) AoT300, (**b**) Att300, (**c**) AoT2500, and (**d**) Att2500. The line plots correspond to the subject average, while shaded areas correspond to the standard deviation.

**Figure 3 sensors-22-01694-f003:**
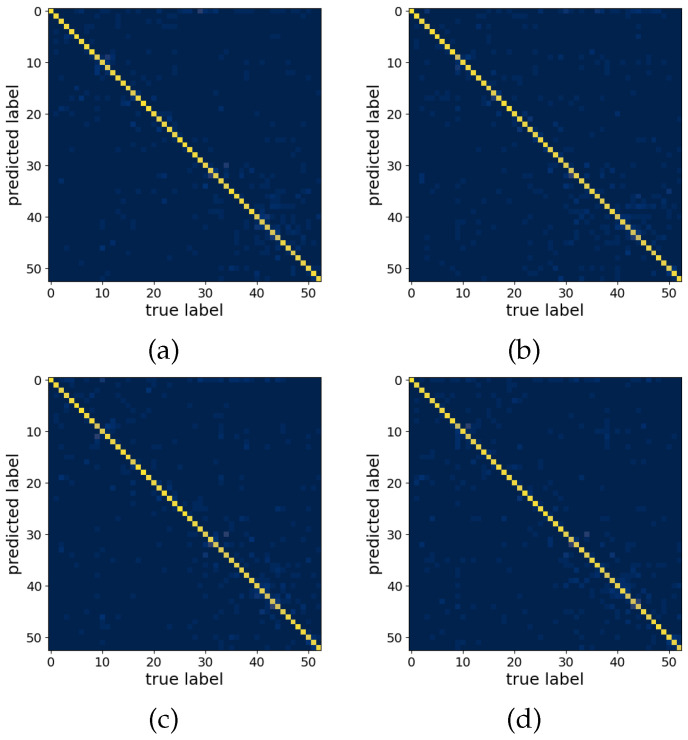
Confusion matrices for each of the models. (**a**) AoT300, (**b**) Att300, (**c**) AoT2500, and (**d**) Att2500.

**Figure 4 sensors-22-01694-f004:**
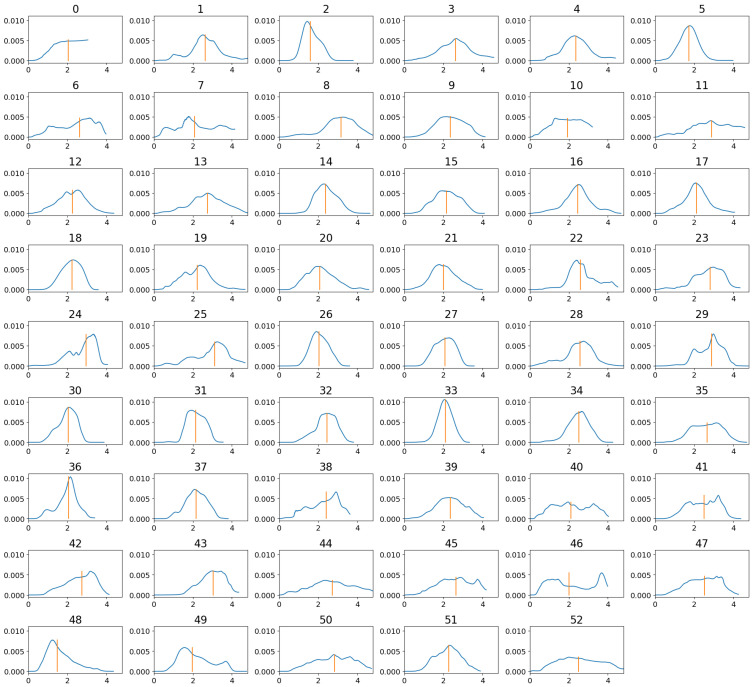
Distribution of attention weights extracted from each gesture (1–52, 0 = resting state) in the training set for the Att300 model. The vertical orange line corresponds to the median, and the x-axis is time in seconds.

**Figure 5 sensors-22-01694-f005:**
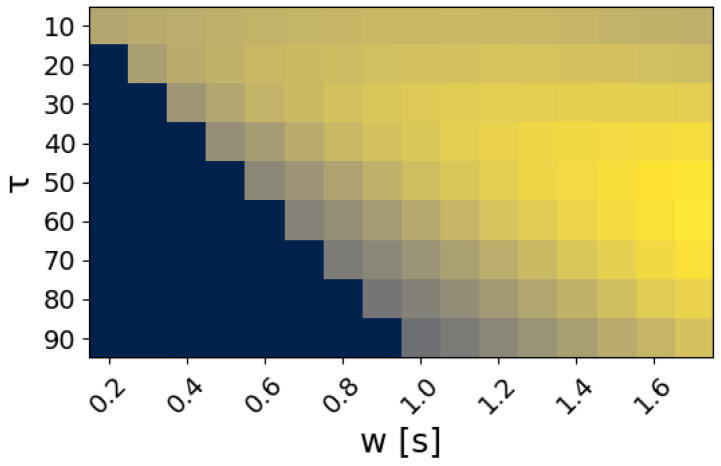
Results of the grid search for the τ and *w* parameters. Colors represent the achieved accuracy from low values (purple) to the maximum (yellow). The highest accuracy is achieved for τ=60 and w=1.6s.

**Figure 6 sensors-22-01694-f006:**
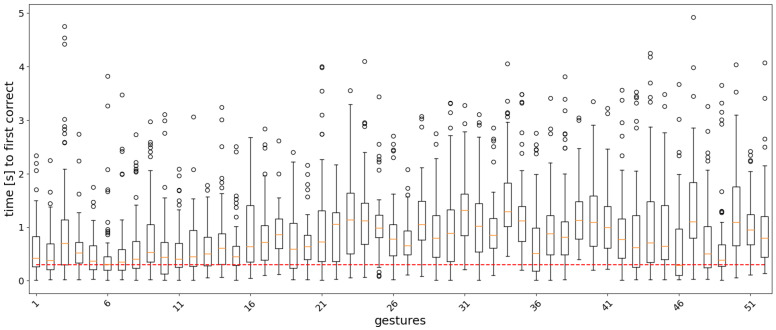
Boxplots of the distribution of the time to the first correct prediction per gesture for the AoT300 model. The boundaries of the boxes correspond to the Q1 and Q3 quartiles, the boundaries of the standard deviations are Q1 − 1.5IQR and Q3 + 1.5IQR, where IQR = Q3 − Q1, and the circles correspond to outliers. The red dashed line corresponds to 300 ms.

**Figure 7 sensors-22-01694-f007:**
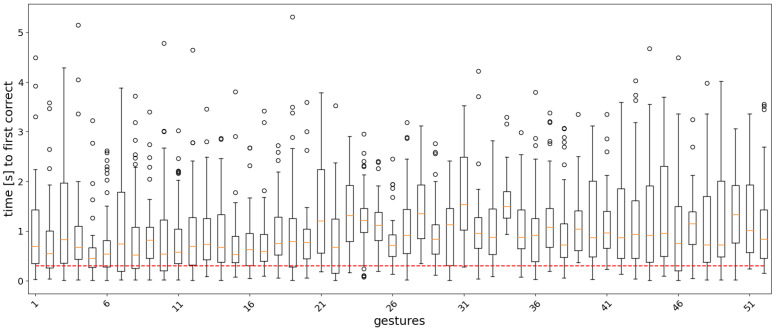
Boxplots of the distribution of the time to the first correct prediction per gesture for the AoT2500 model. The boundaries of the boxes correspond to the Q1 and Q3 quartiles, the boundaries of the standard deviations are Q1 − 1.5IQR and Q3 + 1.5IQR, where IQR = Q3 − Q1, and the circles correspond to outliers. The red dashed line corresponds to 300 ms.

**Figure 8 sensors-22-01694-f008:**
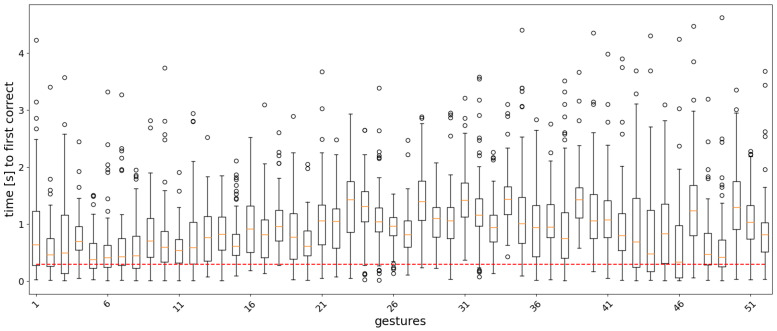
Boxplots of the distribution of the time to the first correct prediction per gesture for the Att300 model. The boundaries of the boxes correspond to the Q1 and Q3 quartiles, the boundaries of the standard deviations are Q1 − 1.5IQR and Q3 + 1.5IQR, where IQR = Q3 − Q1, and the circles correspond to outliers. The red dashed line corresponds to 300 ms.

**Figure 9 sensors-22-01694-f009:**
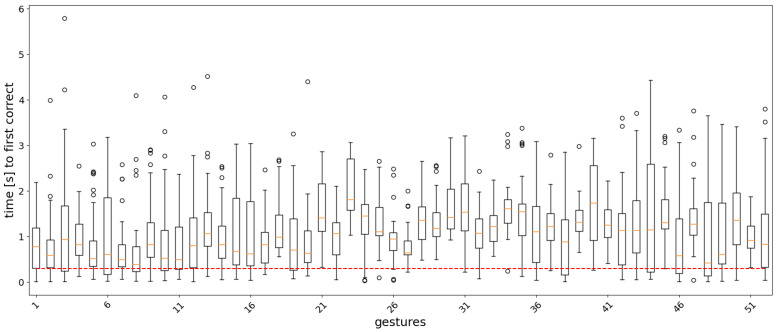
Boxplots of the distribution of the time to the first correct prediction per gesture for the Att2500 model. The boundaries of the boxes correspond to the Q1 and Q3 quartiles, the boundaries of the standard deviations are Q1 − 1.5IQR and Q3 + 1.5IQR, where IQR = Q3 − Q1, and the circles correspond to outliers. The red dashed line corresponds to 300 ms.

**Figure 10 sensors-22-01694-f010:**
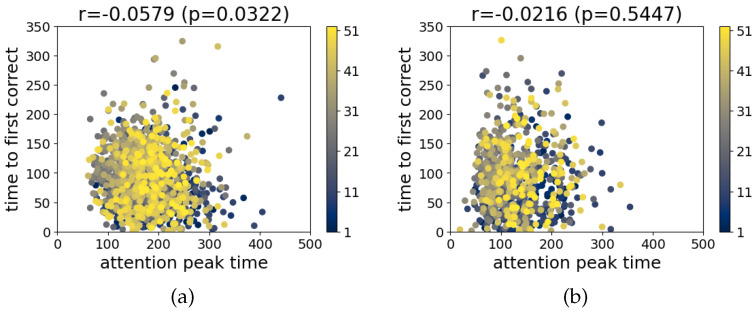
Correlation between attention layer peaks and time to first correct prediction for each of the models. (**a**) Att300 and (**b**) Att2500.

**Figure 11 sensors-22-01694-f011:**
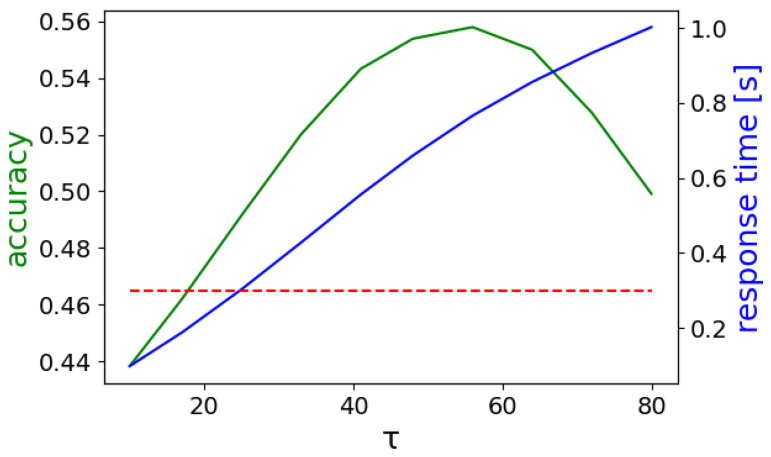
Classification accuracy (green) and response time (blue) when w=1.6 s and τ is varied for the Att300 model. The red dashed line corresponds to 0.3 s.

**Table 1 sensors-22-01694-t001:** Details of the evaluated models, average over time (AoT), and attention mechanism (Att). The number of layers refers only to convolutions.

Classifier	RF [ms]	Size	Layers
AoT	300	60 K	4
AoT	2500	70 K	7
Att	300	75 K	4
Att	2500	85 K	7

**Table 2 sensors-22-01694-t002:** Hyperparameter values for each of the augmentation methods, Wavelet Decomposition (WD), Magnitude Warping (MW), and Gaussian Noise (GN).

Augmentation	Hyperparameters
WD	wavelets = [‘sym4’], levels = [2, 3, 4], b = [0, 2.5, 5], p = 0.75
MW	sigma = 0.2, p = 0.75
GN	snrdb = 30, p = 0.25

**Table 3 sensors-22-01694-t003:** Offline and real-time results in terms of classification accuracy. The table shows the average across subjects and the standard deviation in parentheses. An ‘*’ denotes a significant difference (p-value < 0.05) between the top-1 accuracy in [31] and the current study.

Model	Offline Top-1 Accuracy [31]	Offline Top-1 Accuracy	Offline Top-3 Accuracy	Real-Time Accuracy	Response Time (ms)
AoT300	0.8951 (0.0343)	0.9189 (0.0366) *	0.9832 (0.0157)	0.4293 (0.0415)	122.83 (0.89)
AoT2500	0.8929 (0.0380)	0.9147 (0.0402) *	0.9788 (0.0177)	0.2022 (0.0439)	121.29 (0.84)
Att300	0.8967 (0.0350)	0.9067 (0.0443)	0.9790 (0.0170)	0.4188 (0.0429)	122.51 (0.94)
Att2500	0.8976 (0.0349)	0.9100 (0.0365)	0.9774 (0.0165)	0.1772 (0.0435)	120.76 (1.34)

**Table 4 sensors-22-01694-t004:** Offline and real-time results in terms of classification accuracy when training is performed using a sliding windows of 200 ms. The table shows the average across subjects and the standard deviation in parentheses.

Model	Offline Top-1 Accuracy	Offline Top-3 Accuracy	Real-Time Accuracy	Response Time [ms]
AoT300	0.7442 (0.0548)	0.9019 (0.0349)	0.7527 (0.0582)	118.54 (1.56)
AoT2500	0.7619 (0.0618)	0.9079 (0.0383)	0.7696 (0.0667)	117.61 (1.50)
Att300	0.7062 (0.0531)	0.8825 (0.0314)	0.7481 (0.0630)	120.24 (1.56)
Att2500	0.7800 (0.0528)	0.9169 (0.0314)	0.7867 (0.0561)	119.31 (1.72)

## Data Availability

Publicly available datasets were analyzed in this study. These data can be found here: http://ninapro.hevs.ch/ [Accessed: 10 December 2021]. The code and data generated from this study are openly available at https://github.com/DSIP-UPatras [Accessed: 10 December 2021].

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
