# Peer review of "Real-Time Analysis of Hand Gesture Recognition with Temporal Convolutional Networks†"

_sensors, 2022, doi:10.3390/s22051694_

Round 1

Reviewer 1 Report

The authors utilise the Temporal Convoluntional Networks to classify the EMG signal data, and evaluate the performance. For this manuscript, I have several suggestions.

  1. In the abstract, there is no need to label the sentences with (1)Background (2) Methods, etc.
  2. It's better to represent the formulas out of the paragraph rather than in the sentence.
  3. There should be a figure discribing the whole structure of the TCN model.
  4. Please add more information about how do you use the TCN model and what kind of optimisation have you done.

Author Response

Comment 1: In the abstract, there is no need to label the sentences with (1) Background (2) Methods, etc.

The labeling of the sentences in the abstract was done in order to comply with the MDPI template.

Comment 2: It’s better to represent the formulas out of the paragraph rather than in the sentence.

In the new version, all the equations are on separate paragraphs.

Comment 3: There should be a figure describing the whole structure of the TCN model.

In the paper, Figure 1 already shows the TCN architecture, where Figure 1-b shows the building blocks, i.e., the sequence of layers, the residual connections and the two output versions (AoT and Att), while Figure 1-a shows how the causal convolution with dilations work.

In the revised manuscript, the caption of Figure 1 has been improved to describe better the different components.

Comment 4: Please add more information about how do you use the TCN model and what kind of optimisation have you done.

More information about how the TCN model works has been added (Section 2, ln 117-128), while optimization details are described in Section 2, ln 150-167.

Reviewer 2 Report

In this paper the authors extend their previous research on TCN models changing somehow the experiment configuration and performing real-time analysis.

Some of the paragraphs are copied verbatim from the previous paper, therefore I would suggest to rephrase them.

In line 69 "improved offline accuracy compared to our previous study" you would have to state how this is achieved. What is exactly the difference in the experimentation between this and your previous paper? Just the change of some parameter values? The only difference I could see is that in your previous paper you trained the network for 30 epochs and now for 100. Is there any other change? If not, I would focus the contribution of the paper in the online evaluation and mention the improved offline accuracy in a secondary place. This would also be applicable to the abstract.

You should make it clear that you are not doing real-time analysis with real-time data, but simulating real-time with offline data.

Author Response

Comment 1: Some of the paragraphs are copied verbatim from the previous paper, therefore I would suggest to rephrase them.

The sections of the manuscript that were copied from the original paper have been modified (Section 2, ln 82-160).

Comment 2: In line 69 "improved offline accuracy compared to our previous study" you would have to state how this is achieved. What is exactly the difference in the experimentation between this and your previous paper? Just the change of some parameter values? The only difference I could see is that in your previous paper you trained the network for 30 epochs and now for 100. Is there any other change? If not, I would focus the contribution of the paper in the online evaluation and mention the improved offline accuracy in a secondary place. This would also be applicable to the abstract.

The changes were: training for more epochs and using different augmentation techniques resulted from our previous study [1]. To that effect, an extended hyperparameter search was also applied in order to find the values that finally resulted in the improved offline metrics.

We took into consideration the reviewer’s comment and modified the related sections of the manuscript (Section 1, ln 68-77 and Section 2, ln 157-160).

Comment 3: You should make it clear that you are not doing real-time analysis with real-time data, but simulating real-time with offline data.

We clarified in the abstract (ln 7-8) and the methodology description (Section 2, ln 169-171) that our experimentation is based on simulating the real-time signals with offline data taken from the dataset.

Reviewer 3 Report

The authors described a gesture detection solution based on Temporal Convolutional Networks. In the article, they were based on their previous work, therefore I have a few comments:

1. In the introduction, the authors describe their contribution to this article. It is worth extending this description as it is now very brief.
2. In section 2 they describe the TCN model, but I'm not sure how new this description is to previous work. Please comment and expand this description in the article.
3. In the experimental part, it is worth extending the description of the used databases.
4. Discussion of the research results is sufficient.

Author Response

Comment 1: In the introduction, the authors describe their contribution to this article. It is worth extending this description as it is now very brief.

In the revised manuscript, the description of the contributions of this work in relation to our previous paper has been improved (Section 1, ln 68-77).

Comment 2: In section 2 they describe the TCN model, but I'm not sure how new this description is to previous work. Please comment and expand this description in the article.

In order to differentiate from the original paper, the description of the TCN architecture has been revised (Section 2, ln 82-160).

Comment 3: In the experimental part, it is worth extending the description of the used databases.

In the revised manuscript, more details are provided in the description of the Ninapro database regarding the subjects, the gestures and the recording equipment (Section 2, ln 129-139).